# External Basic Hyperthermia Devices for Preclinical Studies in Small Animals

**DOI:** 10.3390/cancers13184628

**Published:** 2021-09-15

**Authors:** Marjolein I. Priester, Sergio Curto, Gerard C. van Rhoon, Timo L. M. ten Hagen

**Affiliations:** 1Laboratory of Experimental Oncology, Department of Pathology, Erasmus University Medical Center, 3015 GD Rotterdam, The Netherlands; m.priester@erasmusmc.nl; 2Department of Radiotherapy, Erasmus MC Cancer Institute, University Medical Center Rotterdam, 3015 GD Rotterdam, The Netherlands; s.curto@erasmusmc.nl (S.C.); g.c.vanrhoon@erasmusmc.nl (G.C.v.R.)

**Keywords:** hyperthermia, preclinical hyperthermia, small animals, standardization, multimodal therapy

## Abstract

**Simple Summary:**

The application of mild hyperthermia can be beneficial for solid tumor treatment by induction of sublethal effects on a tissue- and cellular level. When designing a hyperthermia experiment, several factors should be taken into consideration. In this review, multiple elementary hyperthermia devices are described in detail to aid standardization of treatment design.

**Abstract:**

Preclinical studies have shown that application of mild hyperthermia (40–43 °C) is a promising adjuvant to solid tumor treatment. To improve preclinical testing, enhance reproducibility, and allow comparison of the obtained results, it is crucial to have standardization of the available methods. Reproducibility of methods in and between research groups on the same techniques is crucial to have a better prediction of the clinical outcome and to improve new treatment strategies (for instance with heat-sensitive nanoparticles). Here we provide a preclinically oriented review on the use and applicability of basic hyperthermia systems available for solid tumor thermal treatment in small animals. The complexity of these techniques ranges from a simple, low-cost water bath approach, irradiation with light or lasers, to advanced ultrasound and capacitive heating devices.

## 1. Introduction

Hyperthermia (HT) is a therapeutic modality in which tissue temperature is elevated above physiological temperature for a predefined period of time. Application of mild hyperthermia (40–43 °C) is able to induce sublethal effects to the target region, which can induce beneficial mechanisms on both a tissue- and cellular level [1,2]. Clinical hyperthermia can be divided into three categories: local-, partial-, or whole body hyperthermia [3]. To study the application, effects in a treatment setting, and mechanisms involved in HT, different methods have been developed. The selection of the most suitable heating method depends on the research question, (pre)clinical setting, and most importantly on the complexity of the heating method needed.

Hyperthermia is usually administered as an adjuvant therapy to improve the therapeutic efficacy of existing radio- and/or chemotherapies. The main focus of this review is devices inducing mild, local temperature elevation in the treatment of small animals bearing solid tumors. Various methods have been developed, varying in their way of energy transfer, heating volume, site of application, complexity, and costs [4]. In the following sections, the general considerations for designing a preclinical heating study are outlined. Subsequently, the available hyperthermia techniques are described in detail. These range from simple setups such as water baths, cold light sources, and near-infrared lasers, to advanced focused ultrasound and capacitive hyperthermia devices. This review provides an outline on devices and adaptations thereof, which can be used for application of mild HT in small animals, and helps to identify the optimal setup depending on the research question.

In this review we present insight in the use and adaptations of hyperthermia setups for small animals. The articles reviewed give adequate insight into the methodology and applicability of the systems. To collect articles for this purpose, we searched PubMed, Web of Science, and Embase databases systematically for articles published before 15 July 2021. The search terms included, amongst others, “hyperthermia”, “(solid) tumor”, “animal”, and the discussed HT technique. Titles and abstracts were initially screened for meeting the inclusion criteria. Full articles were retrieved for all eligible studies and checked for additional material when appropriate. Key publications cited in included articles were also checked.

Two exclusion steps were applied for selection of papers concerning hyperthermia devices in preclinical studies (Figure 1). Firstly, exclusion criteria were as follows: 1; studies on larger animals such as rabbits, canines, sheep, and pigs, 2; studies focusing on HT applications in vitro, ex vivo, or simulations, 3; studies focused on non-oncological treatments, 4; studies solely applying treatment temperatures above mild hyperthermia, 5; reviews, abstract-only conference papers, editorials, and books, 6; any publication in a language other than English, and 7; articles without available full text. In addition to the exclusion criteria, articles were assessed on their design and methods.

Secondly, only original research articles, which presented a detailed description of the methodology, a new design functionally different from previous setups, or an essential update on an existing method, were included. The purpose of this review is to provide insight in the methodologies, different methods available, and the improvements thereupon. Therefore, articles were selected on the basis of detailed description of methodology, and not, per se, on the basis of results obtained with these setups as this falls outside the scope of the review. After selection based on the above exclusion steps, 55 papers remained presenting either the water bath (*n* = 25), cold light source (*n* = 6), near-infrared laser (*n* = 8), high-intensity focused ultrasound (*n* = 9), or capacitive hyperthermia (*n* = 7) methods.

## 2. Study Design—General Considerations

### 2.1. Tumor Model

Animal—While the majority of the preclinical heating studies have been performed in mice, some groups prefer the use of either rats or hamsters. The inter-species variation should be taken into account as various physiological and anatomical factors contribute to differences in sensitivity. This may not only be inconvenient for the comparison of studies, but also hampers translation to the clinic [5,6,7]. Intra-species variation also occurs due to diversity in the genetic background among strains. For instance, a study by Hunter et al. shows that metastatic capacity is dependent on the selected murine strain. While the oncogenic driver was identical in all strains, the inherent variability lead to a 40-fold difference in metastatic burden [8].

Tumor—Solid tumor development is dependent on both the cell type and the tumor micro-environment (TME). The origin of the cell line is a variable factor, as murine and human tumors do not necessarily behave in a similar manner [9]. Some notable characteristics could be differences in growth rate, spatial distribution, vascular development, and stromal components [9]. A typical characteristic during hyperthermia treatment is differential sensitivity, as some malignant cells are more susceptible to heat than others [10,11]. Besides type and number of cancerous cells present, the anatomical site of the tumor is also a critical factor. Deviation from the natural tumor location might affect cell survival and TME remodeling, which could result in either improved or diminished treatment response [12]. The establishment of the tumor into the model, either spontaneous, xenograft, or syngeneic, might also influence the heating efficiency [13,14]. The animal may bear either a single or bilateral tumor. In the case of a single tumor, it is possible to include the non-tumor side in the treatment regimen as a control depending on the study objective [15,16]. In case of a bilateral tumor, one malignancy could receive hyperthermic treatment and the other may act as an unheated intraindividual control. Lastly, tumor volume at the onset of treatment should be considered. While the tumor development time is dependent on the selected cell line, the study objective determines the desired tumor size. In general, for hyperthermia studies, the following tumor size ranges are considered to be experimentally suitable: for mice 50 to 700 mm^3^, for rats 100 to 2200 mm^3^, and for hamsters 50 to 150 mm^3^, as mentioned in the tables below. Overall, all of the described parameters should be considered during study design as these are able to influence the outcome of the experiment.

### 2.2. Anesthesia and Analgesia Management

Anesthetic and analgesic management is applied in preclinical experimental procedures in order to minimize distress and discomfort [17]. Most experimental procedures in small animals are conducted using inhalation anesthesia (isoflurane; induction 3%, maintenance 1–3%) in medical air or pure oxygen (flow 0.4–0.6 L/min). Others use a combination of ketamine (100 mg/kg) and xylazine (10 mg/kg), which is recommended for short procedures as it induces a surgical plane for approximately 30 min [18]. During general anesthesia, application of ophthalmic ointment is recommended to prevent dehydration of the corneal surface. A pain suppressor is generally administered at least 30 min prior to start of the procedure and the dose is repeated once 6 h after initial administration (e.g., Buprenorphine S.C. injection; 0.05 mg/kg). Post-operative analgesia includes administration of a non-steroidal inflammatory drug (NSAID) once per day up to 48 h after surgery (e.g., Carprofen S.C. injection; 3–5 mg/kg) [17,19]. The application of general anesthesia ensures immobilization; however, it is also associated with various side effects related to the relatively small size of rodents. Almost immediately after induction, cardiovascular and respiratory depression may occur and simultaneously the core body temperature will decrease, eventually leading to hypothermia [17,20,21]. Particularly in heating studies, temperature variation within the animal should be kept to a minimum. A study by O’Hara et al. showed that anesthetized animals did not only show a reduction in core temperature, but also a decrease in the intratumoral temperatures in both the control and heated subjects [22]. Therefore, monitoring of vital parameters is essential during animal studies.

### 2.3. Temperature Monitoring

The normal rectal temperature in small animals varies slightly for mice (36.5–38.0 °C), rats (37.5–38.5 °C), and hamsters (37.0–38.0 °C) [23]. The rectal temperature should not rise above 39 °C, nor manifest a maximum variation of 0.5 °C during the heating studies [24,25,26]. Thermoregulation can be maintained during the experiment with the use of heater air, -plates, -pads, or reflective foils [27,28]. Some studies have simultaneously applied cooling of the top surface by gently blowing room temperature air over the animal. For instance, studies applying water bath hyperthermia have incorporated air-cooling to avoid increase of physiological core temperature due to thermal conduction [22,29,30].

Preliminary heating experiments should be performed to determine heat distribution for each system at various settings. In order to obtain the general heating pattern, the temperature should be monitored at multiple locations in the tumor center and -periphery [29,31,32]. In the past, most of the preclinical HT studies measured the intratumoral temperature with the use of needle-type thermocouples. However, the invasiveness of this method should be taken into account, as probe insertion leads to parenchymal disruption, and may create artefacts in tumor circulation [33]. Another factor to take into consideration for temperature monitoring with thermocouples is the potential radiofrequency interaction. Application of capacitive hyperthermia may cause interaction between the conductive metallic components of the thermocouple and the electromagnetic field, resulting in measurement artefacts [34,35]. Additionally, self-heating of the metallic-based thermocouples could eventually cause overestimation of the tissue temperature [36].

In a more advanced setup, hyperthermia profiles could be determined using non-invasive real-time imaging techniques in the form of infrared (IR)- or magnetic resonance (MR) thermometry [37,38]. IR thermometry is able to provide 2D temperature maps, as only superficially emitted infrared energy is converted into temperature profiles [39,40]. Non-invasive 3D thermal monitoring can be obtained using proton resonance frequency shift (PRFS) methods. PRFS is based on the acquisition of phase distribution images through gradient echo (GRE) sequences. Reference-based temperature maps can be generated in real time by subtracting the background image acquired before hyperthermia application from the images obtained during the experiment. However, it is important to consider that this technique only monitors the change in temperature and does not provide absolute temperatures [41]. In addition, both non-invasive imaging techniques may not provide the same accuracy as the thermometry probes due to tissue dependence and susceptibility to motion artifacts [40,42].

## 3. Hyperthermia Devices

### 3.1. Water Bath

Application of hyperthermia by immersion of the target area into a temperature-controlled water bath is a low-complexity technique (Figure 2A). In the past decades, various groups have studied the effect of this approach on mice, rats, and hamsters (Table 1). In order to minimize harmful effects to non-targeted tissue regions, there are factors to consider for standardization in future experiments.

In animal experiments involving the water bath, the animal receives a subcutaneous tumor suspension or fragment most frequently on one or both hind limbs, but water bath heating on the flank or mammary fat pad is also an option (Table 1). Both minor and major precautions aid the protection of healthy tissue surrounding the tumor during heat application. For instance, application of Vaseline cream to the tumor border forms a protective barrier, which protects non-cancerous tissues from heating damage and, therefore, prevents possible skin burns [43,44,45]. Others have covered the to-be immersed leg with a thin plastic bag in order to prevent excessive water absorption. On the long term, this will prevent formation of limb edema [32,46,47]. Another possibility is the use of Tesa™ tape to fix the animal in the correct position. The tail, leg, or body can be loosely taped to a jig or device in order to ensure a similar maximum depth of immersion for every experiment [24,32,33,48]. Some studies have ensured full extension of the immersed leg via attachment of a sinker (approximately 4 g) to the plantar surface of the foot [26,29,33,49].

The animal is subsequently placed on a platform, which should be adapted to reduce thermal conduction from the water bath and simultaneously ensure complete immersion of the tumor-bearing leg. The stage should be covered with an insulating material, such as polystyrene or plastic, in order to minimize an increase in body temperature due to heat radiation [15,26,50,51]. This layer should be thin enough to establish complete immersion of the malignant tissue, as the creation of an air-tissue interface will lead to convective heat loss [22]. Combining these modifications reduces the development of hyperthermia related side effects, such as edema or hemorrhagic necrosis, and allows for complete recovery within 7 days post-treatment [32].

The usefulness of this method lies in its user-friendliness and the fact that any water bath can easily be converted to serve as a hyperthermia device. However, the main drawback of this technique is its range of application, as it is only suitable for models studying superficial (subcutaneous- and intramuscular) tumors. Therefore, the system only mimics the clinical setting for certain cancers, such as melanoma.

An aspect to consider when applying water bath hyperthermia is the non-specific temperature elevation. While other HT devices described in this paper focus on targeted treatment, water bath heating is rather non-specific, affecting an area significantly larger than only the tumor. This creates a different environment, accommodating more than just a solid tumor and the aberrant vasculature. In non-cancerous tissue, the induction of thermal stress results in a thermoregulatory response, which may subsequently result in an increase in local perfusion [52]. Previously, Tungjitkusolmun et al. and Ware et al. observed that dissipation of thermal energy occurs depending on the distance between blood vessels and the tumor [53,54]. As blood flow in larger vessels can act as a heat sink, the femoral vasculature in the water bath model may contribute to thermal efflux [52]. Therefore, temperatures in healthy tissue of the hind limb may rise above physiological temperature during treatment, but as long as mild hyperthermic temperatures are applied no permanent damages are expected to occur. This is in contrast to tumor tissue, as its poor circulation has created a hostile micro-environment due to both structural and functional abnormalities of the vasculature [55]. Therefore, the tumor is unable to respond to thermal stress in a similar manner as healthy tissue. As the heat may not be able to diffuse in order to restore homeostasis, the TME is more vulnerable to heat damage [56].

The application of water bath hyperthermia may also result in non-homogeneous heat distribution. As mentioned in Section 2.3, assessment of tumor temperature gradient can be obtained by placement of invasive tissue probes at various depths relative to the tumor surface. This can be either central (±9 mm), lateral (±6 mm), or peripheral (±3 mm) [41]. As the heat source is located externally, it is plausible that the highest temperature is always measured in the periphery of the tumor and that temperature decreases with increasing tissue depth [41]. As shown by Nishimura et al. and Masunaga et al., temperatures at the tumor center equilibrated within 3–4 min after complete immersion, and remained at 0.2–0.3 °C below the water bath temperature throughout the study [29,30]. The study by Nishimura et al. demonstrated that the depth of immersion is also of importance, as stabilized tumor center temperatures are shown to be 0.1 °C higher at the region most distant from the water surface [29] (Figure 2B). The temperature distribution is dependent on the diameter and depth of the subcutaneous tumor. A study by Willerding et al. showed an increased variation among temperatures at the different probe locations in large tumors exposed to water bath heating in comparison to small tumors [41]. However, the accuracy and precision of probe placement may be low, and temperature values divergent, due to tissue disruption [51]. Therefore, homogeneous heat distribution is thought to occur most optimally in small, shallow tumors located on the distal lower limb or protruding tissue.

Lastly, the integration of either intravital imaging or magnetic resonance imaging (MRI) thermometry could be hampered due to the presence of a large water bolus. Image acquisition is rather impractical as the working distance of the water-dipping lens should attain to the complete immersion depth of the tumor tissue (Figure 2B). MR imaging and -thermometry on the other hand could benefit from presence of the water bolus by homogenizing the magnetic field, called “shimming”. Sumser et al. observed that for hyperthermia purposes, the water bolus fluid should be spiked with Fe_3_O_4_ nanoparticles in order to improve the signal-to-noise ratio [57]. This is in contrast to standard water baths, which frequently use demineralized water.

**Table 1 cancers-13-04628-t001:** Overview of the preclinical studies on application of local hyperthermia using a temperature-controlled water bath sorted by year of publication and animal model.

				Tumor					Device				
Author	Year		Strain	Cell Line	Method	Location	t_Develop_	V_Tumor_ *	TM	HT_D_	HT_T_	HT_t_	Ref.
							Days	mm^3^		°C	°C	min	
Crile	1961	M	Swiss	S180	Fragment	Foot	7	~400	TC	-	41.0–49.0	0–240	[58]
Bleehen	1977	M	BALB/c	EMT6	Suspension	Hind leg	8–10	-	TC	42.0–45.0	40.0–44.0	60	[25]
Stone	1978	M	C3H/Mai	CSU-Mca	Fragment	Hind leg	-	~200	-	42.5–43.0	-	60	[59]
Overgaard	1980	M	C3D2F1/Bom	C3H/Tif	Fragment	Foot	14	200	TC	40.7–45.7	40.5–45.5	60	[60]
Gibbs	1981	M	C3H	C3H	Fragment	Flank/foot	-	~125–300	TC	43.2	41.0–43.0	60	[50]
Rofstad	1982	M	NMRI, Nu/Nu	PDX	Fragment	Hind leg	21–28	200–675	TC	42.5	42.2–42.4	0–180	[61]
Joiner	1982	M	C57BL/Cbi	B16	Suspension	Hind leg	10–14	~350–575	TC	-	43.0	60	[62]
		M	C57BL/Cbi	LLC	Suspension	Hind leg	10–14	~350–575	TC	-	43.0	60	
O’Hara	1985	M	C3H	MCa	Fragment	Hind leg	10–14	~125–300	TC	43.1	42.7	10	[22]
Nishimura	1988	M	C3H/He	MCa	Suspension	Hind thigh	-	500–800	TC	43.0	42.7	10	[29]
		M	C3H/He	SCC VII	Suspension	Hind thigh	-	500–800	TC	43.0	42.7	10	
Cope	1990	M	BALB/c, Nu/Nu	D-54 MG	Suspension	Flank/leg	8	100–200	TC	-	42.0	120/240	[26]
Hauck	1997	M	BALB/c, athymic	D-54 MG	Suspension	Lateral thigh	7–9	200	TC	42.2	41.8	240	[32]
Locke	2005	M	Nu/nu, athymic	HeLa	Suspension	Proximal thigh	-	350	TC	-	41.0/43.0	60	[51]
Peller	2008	M	C57BL/6	BFS-1	Suspension	Hind leg	14	~525	FOP	43.0	-	20–30	[15]
Dicheva	2015	M	C57BL/6	B16BL6	Fragment	Hind leg	-	~50	TC	43.0	42.0	60	[45]
		M	C57BL/6	LLC	Fragment	Hind leg	-	~50	TC	43.0	42.0	60	
Suzuki	1967	R	Donryu	Yoshida	Suspension	Feet	3	180–575	-	40.0–42.0–44.0–46.0	-	30	[63]
Calderwood	1980	R	Wistar	Yoshida	Suspension	Foot	9/16	1000–3500	TC	-	42.0	60	[64]
Dahl	1982	R	BD IX	TCL	Fragment	Thigh	-	-	TC	42.2–45.8	42.0–45.0	0–120	[65]
Wheldon	1982	R	Wistar/CFHB	SSBIa	Fragment	Dorsal		125–400		43.5	43.0–43.3	60	[66]
Van der Zee	1995	R	WAG/Rij	RIOS	Fragment	Thigh	14	~300–1300	TC	43.0	42.0	60	[67]
Moroi	1996	R	Fischer	T9	Suspension	Hind Leg	-	200	TC	43.0	-	30	[68]
Willerding	2016	R	Brown Norway	BN175	Suspension	Hind leg	-	700–2200	TC	41.0–42.0	41.5	60	[41]
Derieppe	2019	R	WAG/Rij	R-1	Fragment	Hind leg	21	~ 1500	FOP	43.0	43.0	10	[69]
Leunig	1992	H	Syrian Golden	A-Mel-3	Suspension	Dorsal S.C.	7	100–150	-	43.0	-	30/60	[70]
Dellian	1993	H	Syrian Golden	A-Mel-3	Suspension	Dorsal S.C.	5	90–140	TC	43.3	-	30	[71]
Pahernik	1999	H	Syrian Golden	A-Mel-3	Suspension	Dorsal S.C.	5	90–140	FOP	37.0/44.0	36.2/43.8	20	[72]

Abbreviation: FOP: fiber optic probe, H: hamster, HT_D_: temperature set on the device, HT_T_: temperature measured intratumorally, HT_t_: hyperthermia treatment time, M: mouse, PDX: patient derived xenograft, R: rat, RIOS: radiation induced osteosarcoma, TC: thermocouple, tDevelop: tumor development time, TM: thermometry, and V_Tumor_: tumor volume. * The tumor volume is converted to mm^3^ if only measurements were provided, using the following formula: V = 0.5 (A·B^2^). Estimates are indicated by “~”.

### 3.2. Cold Light Source (CLS)

Cold light sources are devices that are used to heat tissues, but cause limited superficial heat deposition due to utilization of filtered light, because of which the designation “cold” is used. CLS devices are based on incandescent light bulbs, indicating that the emission of light is caused by heating of the internal filament [73]. A distinction can be made between the use of a simple halogen lamp, or a more advanced water-filtered infrared-A (wIRA) radiation system (Table 2). Standard halogen lamps emit infrared (IR) radiation, which generates heat via a combination of infrared-A (IR-A, λ = 760–1400 nm), infrared-B (IR-B, λ = 1400–3000 nm), and infrared-C (IR-C, λ = 3000 nm–1 mm) [74].

Emission throughout the entire infrared spectrum leads to heat absorption mostly in the superficial skin layers, as this region has the highest water content [75]. Subsequently this may lead to undesired side effects varying from painful sensations to irreversible tissue damage, depending on factors such as treatment duration and intensity [76]. Therapeutically relevant heating can be obtained by the use of a series of band-pass filters in order to emit light only of the desired wavelength [77]. A more sophisticated solution is the addition of a water filter to the setup. This eliminates the presence of both IR-B and IR-C, leading to a higher transmission into the skin and therefore an increase of the maximum tolerable radiation power [74,75,78,79]. The application of unfiltered heat radiation with the use of a standard halogen lamp system (EFR type) is still possible, with the limitation of the maximum intratumoral temperature being set to 41 °C [80]. Exceeding this temperature leads to the aforementioned skin damage [80]. A cold light source, for instance, the KL series from SCHOTT, Mainz, Germany, can easily be adapted for hyperthermic treatment by the removal of the glass filter.

To establish the in vivo tumor model, the animal receives a subcutaneous tumor suspension or fragment on a strategic location on the body (Table 2). The location of the superficial tumor is adaptable and dependent on the study aims, as the motility of the fiber-optic light guide(s) allow(s) for freedom of positioning of both animal and tumor (Figure 3A). In general, the most preferable tumor site is either back, hind leg, or one of the hind feet. Prior to start of the treatment, the animal is anesthetized and subsequently placed into the correct position and immobilized using elastic adhesive bandage (Vet Wrap^®^) [28]. In order to minimize heat application to the surrounding tissues, tumor boundaries are masked using white cotton wool, surgical swabs, or polystyrene [41,80,81]. However, it should be noted that gauze-style dressings do not completely avoid IR radiation as diffuse transmission still occurs through the weave pattern [82]. Tumor tissue is heated using a fiberoptic light source. The design of the applicator consists of a flexible light guide, which facilitates the exact placement of the light beam on the tumor area (Figure 3B). The use of multiple guides ensures a uniform heating pattern in larger tumors [41,80]. The intensity of the lamp can be adjusted to the desired input using an adjustable power source (variac) [81,83] (Figure 3C).

The use of CLS for the application of local hyperthermia shows promise due to the ease of assembly, steering, and alteration of intensity. However, only a limited number of groups have published this technique in the past decades (Table 2), including experimental tumor treatment in larger animals such as piglets and equines (not included in this overview) [84,85]. Hyperthermia using incandescent light can be applied to a more defined target region in comparison with full immersion in a water bath, but at the same time the application of this technique is still limited to superficial skin malignancies. Upon penetration into the skin, wave attenuation occurs due to absorption and scattering. In the case of unfiltered heat radiation, the penetration depth does not exceed several millimeters. This results in the formation of a heat gradient, with high absorption in superficial layers and deposition of sub-cytotoxic heat levels at deeper tissue regions [86]. However, it is important to realize that the cold light source profile follows black-body radiation. This means that temperature is inversely correlated with the wavelength of emitted light, for example, emitted light changes from a faint red glow to white, and, finally, into a blue color with increasing temperature [87,88].

In order to generate homogeneous heating in large tumors, application of mild hyperthermia using wIRA was suggested [89]. Previous studies have observed that a change in wavelength, from 500 to 700 nm, leads to a significant reduction in optical absorbance. Therefore, penetration depth will be enhanced, resulting in therapeutically relevant tissue heating up to a depth of 5 mm [76]. Radiative penetration reaches depths up to 10 mm for both halogen and wIRA systems; however, these may be subtherapeutic heating levels.

**Figure 3 cancers-13-04628-f003:**
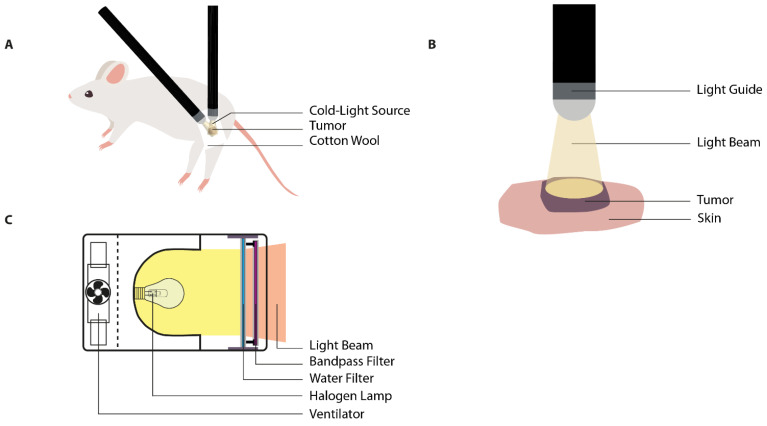
Illustrations of the cold light source for hyperthermia application. (**A**) The heat application through one or multiple flexible fiber optic light guides allows for relatively free positioning of the subcutaneous tumor location. The healthy tissue can be shielded from illumination by placement of, for instance, cotton wool; (**B**) the light source is placed above the target area without interference of any medium. The geometry of heat application is dependent on the width of the light beam and the distance of the light source in relation to the tumor; (**C**) the water-filtered infrared-A (wIRA) radiation system eliminates the presence of both IR-B and IR-C due to the water filter. This provides a light beam with a high penetration depth. Image C is adapted from Kelleher et al. and Vaupel et al. [76,90].

**Table 2 cancers-13-04628-t002:** Overview of the preclinical studies on application of local hyperthermia using a cold light source sorted by year of publication and animal model. A distinction is made between the halogen- and wIRA-based CLS systems.

				Tumor					Device					
Author	Year		Strain	Cell Line	Method	Location	t_Develop_	V_Tumor_ *	TM	λ	Intensity	HT_T_	HT_t_	Ref.
							Days	mm^3^		nm	W/cm^2^	°C	min	
Halogen														
Ickenstein	2003	M	Rag2-M	MDA435	Suspension	Dorsal S.C.	-	20–30	TC	-	-	41.0	60	[91]
Foxley	2012	M	Nu/nu	AT6	Suspension	Hind leg	-	250	TC	-	-	41.0	15	[28]
Sien	1980	R	Sprague-Dawley	W256	Suspension	Back	7	-	THM	-	-	38.0	60	[81]
Limmer	2014	R	Brown Norway	BN175	Fragment	Hind leg	-	~50	TC	-	-	41.0	60	[80]
Willerding	2016	R	Brown Norway	BN175	Suspension	Hind leg	-	700–2200	TC	350–700	-	41.0	60	[41]
wIRA														
Kelleher	1999	R	Sprague-Dawley	DS	Suspension	Hind foot	6	-	TC	665–800	0.80	43.0	60	[90]

Abbreviation: HT_D_: temperature set on the device, HT_T_: temperature measured intratumorally, HT_t_: hyperthermia treatment time, M: mouse, R: rat, tDevelop: tumor development time, TC: thermocouple, TM: thermometry, THM: thermistor, and V_Tumor_: tumor volume. * The tumor volume is converted to mm^3^ if only measurements were provided, using the following formula: V = 0.5 (A·B^2^). Estimates are indicated by “~”.

### 3.3. Near-Infrared (NIR) Laser Light

Laser hyperthermia is based on the conversion of electrical energy into light energy, which is subsequently able to interact with tissues to produce heat [92]. The light emitted by lasers is monochromatic, directional, and coherent. It can be produced in specific wavelengths, varying from visible- to infrared light (λ = 400–3000 nm) [93]. These properties define the extent of tissue penetration and result in predictable temperature profiles [28]. Near-infrared (NIR) frequencies (λ = 750–3000 nm) are able to deliver laser light with high spatial precision over long distances without significant loss of energy [92,94]. As with the cold-light source, there is a window for optimal tissue penetration, which is based on the effective attenuation length. While light absorption depends on various cell- and tissue characteristics, wavelengths of approximately 800 nm are most frequently applied in preclinical studies (Table 3). Showing a penetration depth of up to 10 mm, with minimized excitation of biomolecules and water heating [95,96,97]. However, it should be taken into account that hemoglobin is a dominant background absorber within the 700–900 nm range [98]. This is especially relevant for well-vascularized tumors or malignancies located near major blood vessels, as treatment planning may require parameter alteration to account for heat dissipation. Wavelengths below 700 nm show diminished penetration, while water absorption is more pronounced at wavelengths exceeding 1000 nm [95,96]. The wavelengths selected for preclinical laser-based heating fall between 763 and 1064 nm (Table 3).

Another factor to take into consideration is the Gaussian intensity distribution of the laser beam [99]. When applying laser light using a bare-tipped optical fiber, the directional beam may create a hot spot in the center of the target area. Therefore, there may be non-homogeneous heating, as the tissue located at the boundaries of the beam receives a lower light dose in comparison with tissue in the beam center. Manipulation of the laser beam to establish focalized treatment occurs either through the use of multiple lenses and prisms or with the use of a custom-built illuminator. The diameter of the laser beam can be adjusted to the tumor size by varying the distance between the lenses and the prism [41,100] (Figure 4C). The use of an illuminator provides homogeneous light distribution by passing light through three aligned chambers lined with reflective material (Figure 4B). The size of the beam hitting the tissue surface depends on the size of the exit port in the middle chamber [101,102,103,104]. Upon reaching the tumor tissue, continuous heating occurs as the laser beam will initially heat the upper layer, which gradually spreads to the underlying tissue layers 2–10 mm deep [95,96,97]. In order to establish mild hyperthermia, the output energy of the laser can be adapted by either changing the beam size (mm) or the laser intensity (W/cm^2^).

In animal experiments involving the NIR laser light, the animal receives a subcutaneous tumor suspension or fragment, commonly in the flank, mammary fat pad, or hind limb (Table 3). After application of anesthesia, the treatment site is prepared by shaving and subsequent cleaning with antiseptics. It is possible to opt for either systemic or topical analgesia to alleviate any pain response due to the stimulus [105]. The animal is placed on the stage in an optimal position and immobilized (Figure 4A). During laser-light treatment, both power (W/cm^2^) and pulse interval (s) were manually adjusted in order to ensure target temperature with minimal fluctuations. As mentioned in the general considerations, a non-invasive method to determine the real-time hyperthermia profiles is MRT. After reaching steady state temperatures, the average tumor surface temperature can be monitored during laser light irradiation. In order to ensure MR compatibility, studies have designed holders, mounts and other features based on non-magnetic materials such as polyethylene and glass [41,100,106].

Similarly to CLS, mild thermal treatment via NIR laser light is mainly suitable for superficial tumors [107]. On one hand, the laser system is more advanced than CLS. Laser light is monochromatic and coherent, facilitating homogeneous illumination of the region of interest with distinct borders. However, on the other hand, the laser provides a high-energy light beam which, if not monitored properly, quickly leads to tissue dehydration and charring. This could in the long term lead to inhibition of energy delivery and side effects [106]. Therefore, it is essential to optimize the experimental settings, taking wavelength, beam size, and laser intensity into consideration.

**Table 3 cancers-13-04628-t003:** Overview of the preclinical studies on application of local hyperthermia using NIR laser light sorted by year of publication and animal model.

				Tumor					Device						
Author	Year		Strain	Cell Line	Method	Location	t_Develop_	V_Tumor_ *	TM	λ	Beam	Intensity	HT_T_	HT_t_	Ref.
							Days	mm^3^		nm	ø mm	W/cm^2^	°C	min	
Waldow	1988	M	DBA/2J	SMT-F	Fragment	Axillary region	7–10	120–675	TC	1064	15–20	0.08–0.19	41.0–46.0	25	[105]
Barnes	2013	M	SCK	A/J	Suspension	Hind leg		~30–200		755	8	-	42.5	60	[108]
			SCCVII	C3H	Suspension	Hind Leg		~30–200		755	8	-	42.5	60	
Kirui	2013	M	Nu/nu	CAPAN-1	Suspension	Flank	5–7	290–350	IRC	810	4	1.00	42.0	20	[109]
Zhou	2013	M	Nu/nu	HCT116	Suspension	Dorsal	-	~30–80	IRC	808	-	0.5	31.2–48.8	5	[110]
Dou	2014	M	SCID	ME-180	Fragment	Hind limb	-	~80–300	MRT	763	10	0.50–1.70	42.0	25	[102]
Dou	2015	M	SCID	ME-180	Fragment	Hind limb	14–21	-	MRT	763	10	0.10–0.80	42.0	5	[101]
Panjehpour	1991	R	Sprague-Dawley	Spontaneous	-	MFP	<180	~1650	TC	1064	15	2.40	43.2–43.5	60	[99]
Willerding	2016	R	Brown Norway	BN175	Suspension	Hind leg	9–13	700–1000	TC	940	-	1.00–3.00	>40.0	60	[41]

Abbreviation: FOP: fiber optic probe, HT_D_: temperature set on the device, HT_T_: temperature measured intratumorally, HT_t_: hyperthermia treatment time, IRC: infrared camera M: mouse, MFP: mammary fat pad, MRT: magnetic resonance thermometry, R: rat, tDevelop: tumor development time, TC: thermocouple, TM: thermometry, and V_Tumor_: tumor volume. * The tumor volume is converted to mm^3^ if only measurements were provided, using the following formula: V = 0.5 (A·B^2^). Estimates are indicated by “~”.

### 3.4. Focused Ultrasound (FUS)

Focused ultrasound (FUS), which includes high-intensity focused ultrasound (HIFU), is a non-invasive technique based on local heat delivery to deep tumors via ultrasound waves [42,111] (Figure 5). Energy can be precisely deposited to the target area with minimal damage to intervening and surrounding soft tissues [42,112]. The only prerequisite is the presence of a fairly uniform medium (target tissue), as interference with either air or bone causes acoustic reflections or energy absorption. This could lead to non-uniform heat distribution and subsequent formation of undesired hot spots [42,78,113,114].

The FUS device consists of an imaging transducer for tumor identification and a therapeutic transducer for application of thermal treatment [115]. The ultrasound imaging probe can either be incorporated within the central opening of the FUS transducer or placed directly adjacent to it [116,117]. A distinction can be made between linear- and phased-array imaging transducers [118]. The linear transducer produces a rectangular beam shape, while the phased transducer is able to cover a larger area by either creating multiple focal points or steering a single, large focal zone in a specific direction [115,119]. For the therapeutic transducer, a distinction can be made between the single- and split-beam profiles. Transducers applying split beam technology are able to create a focal zone with a larger symmetrical volume (e.g., 1.49 mm^2^ single focus vs. 7.4 mm^2^ split focus), therefore directly reducing the number of ultrasound exposures required for treatment [42,120]. As demonstrated by Seip et al., application of asymmetric FUS geometries results in a lower maximum tissue temperature. However, this will, in turn, lead to a reduction in cooling time between subsequent sonications, resulting in treatment of a larger lesion volume in one treatment session (Figure 5B) [121]. In order to minimize interference between the FUS beams and the ultrasound imaging, both transducers should not be operated simultaneously. US imaging is performed during the “OFF” period of the FUS pulse with an interval of ~100 μs every 100 ms (Figure 5C) [122].

Several factors should be taken into consideration for optimization of the insonification of the target area. First, depending on the set acoustic intensity FUS can be used for both ablation (high-intensity, >5 W/cm^2^) and hyperthermia (low intensity, <3 W/cm^2^) [42]. Second, the frequency (f) determines the depth of penetration. High-frequency pulses (1–5 MHz) tend to have a shallower penetration depth due to increased attenuation [123,124]. Third, the set duty cycle (DC) determines whether the ultrasound wave is in continuous or pulsed mode. Operating low duty cycles results in decreased temporal average intensities, and therefore generation of non-lethal temperature elevations [119,120]. Fourth, the applied thermal dose is of importance, which is dependent on applied maximum temperature and treatment duration. It should be taken into account that hyperthermia-mediated effects, such as enhanced extravasation, can last up to 24 h in all heating methods [125].

To realize the in vivo tumor model, the animal receives a subcutaneous tumor suspension or fragment, commonly either in the flank, mammary fat pad, or hind limb (Table 4). For instance, Chae et al. opted for tumor formation in the lateral part of the thigh to minimize the influence of breathing motions during FUS therapy [126]. Optimal ultrasound propagation from transducer to tumor occurs in the presence of a coupling medium. This should be a degassed liquid, either water, saline, or ultrasound gel, located in the gap between the FUS transducer and the target tissue [112,127,128]. The construction of a multi-layered interface, which can be composed of ultrasound gel, agar gel, castor oil, mineral oil, or silicon oil, should even further decrease the intensity of backscattering [129]. Another possibility is the use of a coupling cone, which holds both the therapeutic transducer and the coupling medium [112,116,117]. In these systems, the depth of focus (5–50 mm) is determined by the configuration of the cone as both the distance between the transducer and the exit plane and the water pressure can be varied [112,116]. The pressure of the liquid surrounding the HIFU transducer determines the transmission speed of the ultrasound waves. As the transmitted waves have to travel through the coupling cone filled with degassed water, liquid pressure within the cone influences the depth of focus. It is optional to cover the aperture of the cone with a membrane film; however, coupling should still be ensured with the use of a coupling medium between the coupling cone and the tumor surface [112,117]. Another measure performed to improve acoustic coupling is removal of the animal’s fur. A combination of shaving and application of depilatory cream prior to treatment minimizes interference of the ultrasound waves as it is susceptible to the presence of air [112,130].

The FUS therapeutic transducer transmits focused acoustic energy, which forms an ultrasound beam path through the coupling medium. Upon reaching the tumor surface, it produces a focal spot or treatment cell, which should cover the tumor and its margins [42,131]. The shape of the treatment cell is dependent on the geometry of the FUS transducer, frequently resembling a cigar-shaped ellipsoid (width: 1.5–2.0 mm; length: 1.5–2.0 cm) [42]. While a part of the energy will be either absorbed or reflected backwards, there is the possibility that the energy may penetrate deeper than the targeted area. To avoid this, a tumor may be divided into multiple treatment cells. Another option to prevent far-field heating due to acoustic reflections, or propagation into the contralateral thigh, is to opt for either a polyurethane rubber acoustic absorber or a saline bag to control the exit of the ultrasound wave. It is recommended to cover the absorber with a thin layer of isolating material to avoid indirect skin damage due to heat absorbance [114,132,133,134]. Depending on the tumor location and the configuration of the HIFU system, the animal is placed on a platform in an optimal position and immobilized (Figure 5A). Prior to therapeutic treatment, low-power test sonications can be performed in order to correct for focus point aberration [131,132,134,135]. Real-time hyperthermia profiles are determined with either an IR camera, thermocouples, or more commonly with MR thermometry (Table 4).

**Figure 5 cancers-13-04628-f005:**
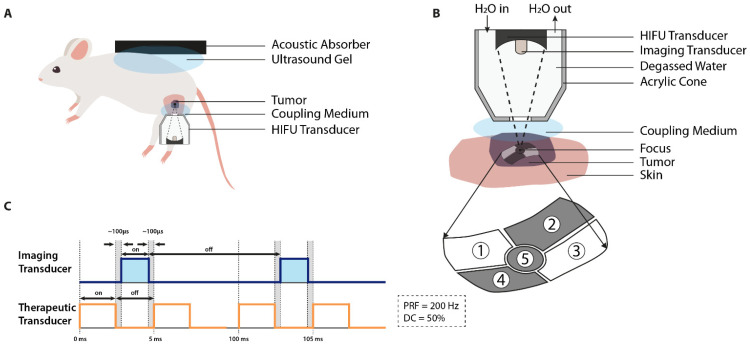
Illustrations of the high-intensity focused ultrasound (HIFU) device for hyperthermia application. (**A**) Both the entrance and the exit of the ultrasound wave should be guided by a coupling medium. At the interface between the HIFU transducer and the tumor, the coupling medium ensures acoustic wave propagation. Additionally, at the other side, where the wave exits the tissue, the presence of a medium in combination with an acoustic absorber prevents far field heating within the animal due to energy reflection; (**B**) the HIFU coupling cone holds both the transducer(s) and the coupling medium. Together, the cone configuration and the coupling medium determines the depth of focus. Incorporation of a split beam transducer allows for application of various heating geometries (element 1–5); (**C**) the sequence control for the therapeutic and imaging transducer. Acoustic interference between the transducers is prevented by non-simultaneous operation. Ultrasound imaging is performed during the “OFF” period of the therapeutic transducer with an interval of ~100 μs. Image B and C are adapted from Wu et al., Seip et al., Lee et al., and Farr et al. [117,121,122,134].

**Table 4 cancers-13-04628-t004:** Overview of the preclinical studies on application of local hyperthermia using focused ultrasound (FUS) sorted by year of publication and animal model.

				Tumor					Device								
Author	Year		Strain	Cell Line	Methods	Location	t_Develop_	V_Tumor_ *	TM	f	TAP	INT	DC	PRF	HT_T_	HT_t_	Ref.
							Days	mm^3^		MHz	W	W/cm^2^	%	Hz	°C	min	
Frenkel	2006	M	BALB/c	JC	Suspension	Flank	18–21	400–700	TC	1.00	20.5	124	9	1.0	41.5	2	[136]
		M	C3H	SCC7	Suspension	Flank	7–10	400–700	TC	1.00	20.5	124	9	1.0	41.5	2	[136]
Patel	2008	M	C3H	SCC7	Suspension	Flank	7–10	700–800	IRC/TC	1.00	20–80	-	10–50	1.0	42	2	[120]
Kheirolomoom	2013	M	FVB	NDL	Fragment	MFP	14	30	TC	1.54	-	-	-	-	42	25	[137]
Chae	2014	M	BALB/c	SCC7	Suspension	Thigh				1.00	12	-	50	5.0	42.9	40	[126]
Cha	2016	M	BALB/c	EMT6	Suspension	Thigh	7–9	150–200	TC	1.50	20	-	15	15.0	42.0	40	[128]
Farr	2017	M	KPC	Spontaneous	-	Pancreas	-	400	MRT	1.20	7	-	-	-	42.5	15	[134]
Centelles	2018	M	SHO	IGROV-1	Suspension	Flank	14	~50–80	TC	1.30	10–20		99.9	1.30	42.0	3–5	[138]
Jeong	2016	M	BALB/c	CT26	Suspension	Dorsal	11	473		1.5	10	84	10	10	42.0	<4	[139]
Hijnen	2012	R	Fisher 344	GS 9L	Suspension	Hind leg	-	400	MRT	1.44	8	117	-	-	42	15	[132]

Abbreviation: DC: duty cycle, HT_T_: temperature measured intratumorally, HT_t_: hyperthermia treatment time, INT: intensity, IRC: infrared camera M: mouse, MFP: mammary fat pad, MRT: magnetic resonance thermometry, PRF: pulse repetition frequency, R: rat, TAP: acoustic power, TC: thermocouple, tDevelop: tumor development time, TM: thermometry, and V_Tumor_: tumor volume. * The tumor volume (V_tumor_) is converted to mm^3^ if only measurements were provided, using the following formula: V = 0.5 (A·B^2^). Estimates are indicated by “~”.

### 3.5. Capacitive Hyperthermia

Capacitive heating is based on generation of an electromagnetic field (8.00 or 13.56 MHz) applied by positioning the tumor-bearing part of the animal between two electrodes, with water or another substance at the tissue-electrode interface to guide the current through the animal. The amount of energy absorbed is dependent on electrical conductivity of the different tissues. It is postulated that absorption is higher in tumors, due to the Warburg effect-related increase in conductivity [140]. Besides by a standard continuous sinusoidal electromagnetic field, the energy can also be applied as modulated electro-hyperthermia (mEHT). mEHT is based on generation of an amplitude-modulated electromagnetic field, which is postulated to generate tumor-specific frequencies [40].

The capacitive setup consists of two plan-parallel electrodes. The lower electrode is frequently incorporated in the grounded, aluminum housing of the stage (Figure 6A). Upon positioning the animal on the stage, there is indirect contact with the lower, static electrode. The stage is temperature-controlled, and connected to the capacitive system with heating and radiofrequency cables [141,142,143,144]. The upper electrode can either be a tissue- or pole electrode (Figure 6B). A tissue electrode (TE) is a small, flexible circular element (Figure 6C). It consists of a fabric cover ring, which encompasses a fabric with a conductive coating, such a copper-silver-tin [141,144]. The TEs are positioned over the tissue by rubber bands, which are stretched and subsequently anchored to the table [141,143,144]. As discussed by Danics et al. (2020), TEs could be inconvenient due to their large size and lack of adaptation to uneven surfaces [141]. Therefore, technical improvements have led to the design of a pole electrode (PE) (Figure 6D). It consists of a column-shaped plastic casing, which houses multiple stainless-steel rods. At the tissue-electrode interface, the device is coated with a conductive fabric, such as silver-plated textile. In comparison to the tissue electrode, PE has a smaller conductive area and improved freedom of both placement and motility. This resulted in a more accurate tissue-electrode contact at curved areas, such as the inguinal region, which improves both focus and coupling [141].

In animal experiments involving capacitive systems, the animal receives a subcutaneous tumor either via suspension or fragment. Common tumor locations are the mammary fat pad, prostate, or hind limb (Table 5). To maximize coupling at the tissue-electrode interface and to minimize the induction of eddy currents, the treatment area benefits from shaving and the subsequent application of a liquid. This could be either ultrasound gel [143], conducting electrode cream [145], or a thin water bolus [146]. Another possibility is the placement of a thin solid insulator at the interface, such as cellophane [141]. Cooling of the upper electrode could be realized using dampened gauze on the TE [143] or incorporation of Peltier elements in the PE [146]. Unlike other hyperthermia devices, external shielding is not required to prevent dispersion as the current flows through a closed, directional electric circuit [147].

Capacitive devices can be applied to various anatomical sites to target both superficial and deep-seated tumors, depending on the diameter of the opposed electrodes [148]. However, the amount of subcutaneous adipose tissue at the target site should be taken into account. As the electric field has to pass through this high-resistance layer, the absorbed energy causes substantial heating [73,149,150]. Upon establishment of the electric field, a current will flow through the target area from one electrode to the other. Being a closed, directional circuit, the formation of undesired hot spots outside the electric flow is limited [151]. Upon passing through the dielectric media (tissue), there will be differences in the permittivity and conductivity [40]. The electromagnetic current will pass through surfaces with a low impedance (high conductivity) as it will by default choose the path of least resistance.

Other types of electromagnetic heating, such as radiofrequency and microwave, have also been employed for preclinical hyperthermia studies [37,152,153]. Radiofrequency-based hyperthermia (3 Hz to 300 MHz) is most frequently used for deep-seated tumors, while microwave heating (300 MHz to 300 GHz) is suitable for treatment of superficial malignancies [154]. Both types of electromagnetic heating may be used for advanced drug delivery studies in combination with intravital microscopy or MRI for real-time visualization. These techniques are discussed in detail by Priester et al., see elsewhere in this special issue.

**Figure 6 cancers-13-04628-f006:**
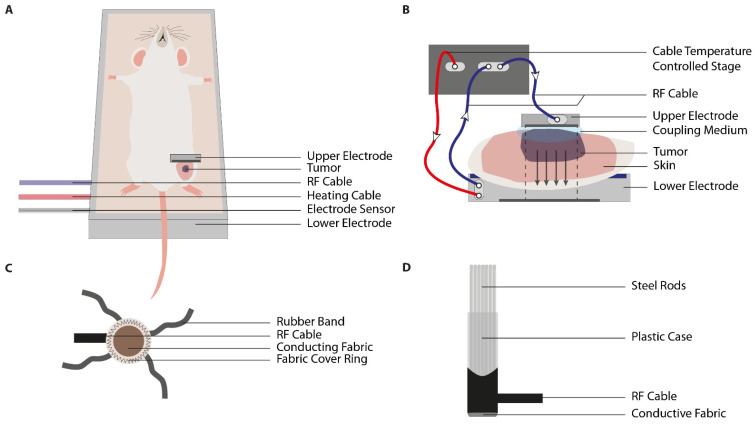
Illustrations of the capacitive hyperthermia device. (**A**) The positioning of the animal on the capacitive set up; (**B**) the lower and upper electrode should be placed plan-parallel for optimal heating. The upper electrode can either be a tissue electrode or a pole electrode; (**C**) the tissue electrode consists of a conducive fabric surrounded by a fabric cover ring. Placement of the tissue electrode is dependent on the use of rubber bands; (**D**) the pole electrode consists of multiple steel rods placed in a plastic casing. The bottom of the plastic casing is lined with a conductive fabric. Positioning of the pole electrode is facilitated by a height-adjustable arm. Figure 6 is adapted from Danics et al., Schvarcz et al., Vancsik et al., and Szasz et al. [141,142,144,155].

**Table 5 cancers-13-04628-t005:** Overview of the preclinical studies on application of local hyperthermia using capacitive hyperthermia sorted by year of publication and animal model.

				Tumor					Device							
Author	Year		Strain	Cell Line	Methods	Location	t_Develop_	V_Tumor_ *	TM	Type	UPS	f	P	HT_T_	HT_t_	Ref.
							Days	mm^3^			ø cm	MHz	W	°C	min	
Danics	2020	M	BALB/c	4T1	Suspension	MFP	7	-	FOP	LabEHY 100/200	2.5/1.8	13.56	0.7 ± 0.3	41.5	35	[141]
Cohen	2019	M	Fox1^nu^	PC3-Luc	Suspension	Prostate	-	150–250	FOP	LabEHY 100	2.5	13.56	0.3–1.0	41.0	30	[143]
Vancsik	2018	M	BALB/c	C26	Suspension	Thigh	14	~900	FOP	LabEHY	2.5	13.56	1.0–3.0	42.0	30	[144]
Andocs	2009	M	BALB/c	HT29	Suspension	Thigh	18	500–800	FOP	LabEHY	2.5	13.56	4.0	42.0	30	[156]
Uchibayashi	1994	M	BALB/c	KK-47	Suspension	Dorsal	<14	200–300	TC	Thermotron RF 8	-	8.00	-	42.5	30	[157]
Marmor	1977	M	BALB/c	KHJJ	Suspension	Flank	10–12	100	THM	Critical Systems	1.2	13.56	0.6–0.9	43.0	30	[145]
			BALB/c	EMT-6	Suspension	Flank	10–12	100	THM	Critical Systems	1.2	13.56	0.6–0.9	43.0	30	[145]
		R						~			2.0					
Shinkai	2002	R	F344	T9	Suspension	Thigh	11	~675–1300	TC	Thermotron RF-IV	2.0	8.00	40.0–60.0	41.0	20	[150]

Abbreviation: F: frequency, FOP: fiberoptic probe, HT_T_: temperature measured intratumorally, HT_t_: hyperthermia treatment time, M: mouse, MFP: mammary fat pad, P: power R: rat, TC: thermocouple, tDevelop: tumor development time, THM: thermistor, TM: thermometry, UPS: upper probe size, and V_Tumor_: tumor volume. * The tumor volume (V_tumor_) is converted to mm^3^ if only measurements were provided, using the following formula: V = 0.5 (A·B^2^). Estimates are indicated by “~”.

## 4. Conclusions

The standardization of mild hyperthermia represents further improvement of preclinical testing, as some attribute disappointing clinical trial results to the lack of supporting preclinical data [158]. This review provides a preclinically oriented detailed overview on systems for solid tumor hyperthermia treatment in small animals (Table 6). By discussing only original research articles, which present a detailed description, a new design, or an essential update on an existing method, standardization is facilitated by addressing both general considerations and method-specific details.

The devices described in this review can be ranked based on technological complexity. Whereas water bath heating is considered to be a low-complexity, entry level device, it is an accurate system for homogeneous yet non-specific heating. Another advantage of this technique is the possibility to simultaneously treat multiple animals, thereby minimizing variation due to exposure to the same conditions [58]. Therefore, the water bath method is suitable to efficiently determine the effects of localized mild hyperthermia. However, it is only a small step progressing from a water bath to a CLS device with distinct advantages. The simple CLS option in the form of a halogen-based device can easily be obtained by making small alterations to an existing light source. The inclusion of a water filter may even further optimize the device, but both are an improvement considering penetration depth, target specificity, and device tuning.

The use of a NIR laser-based device even further enhances HT application, due to an increased penetration depth, a more precise focal zone, and the tuneability of multiple factors. However, thorough optimization of the device is necessary in order to avoid side-effects such as irreversible skin damages. FUS is the most advanced technique in this group, and the versatility allows for thermal treatment of both superficial- and deep tumors. Comparable to the NIR laser-based system, optimization of several settings is required. However, in the end, FUS is a more advanced system with favorable penetration depth, high target specificity, and the possibility to include either a diagnostic transducer or MR acquisition for spatial control and thermometry. However, preclinical studies on FUS are mainly performed in large animals (Figure 1). Similar to FUS, capacitive hyperthermia is a non-invasive technique, which can be used for thermal treatment of both superficial and deep-seated tumors. Capacitive heating takes advantage of tissue dielectric properties, resulting in (non)-thermal therapeutic effects at the target site. However, it should be taken into account that adipose tissue, being a thermal insulator, should be circumvented to avoid substantial heating.

Taken together, the described basic hyperthermia devices provide an excellent foundation for preclinical research when carefully considering the benefits and limitations. The advantage of more advanced devices for certain applications may be worthwhile, however this advancement obviously results in a more demanding, complicated, and costly setup.

## Figures and Tables

**Figure 1 cancers-13-04628-f001:**
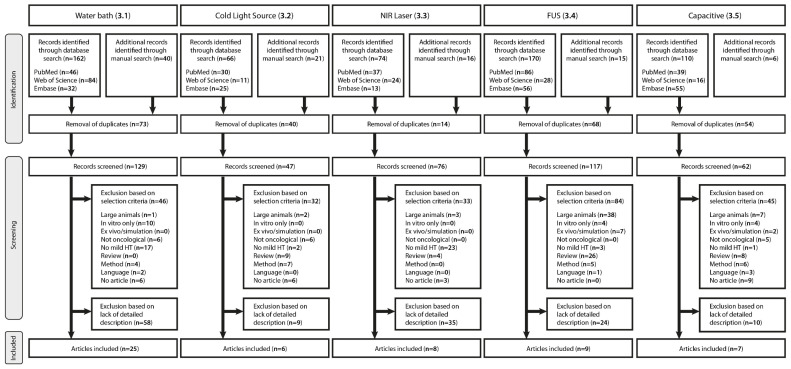
PRISMA flow diagram illustrating selection of the articles for each hyperthermia technique. This review presents a detailed description of the water bath (Section 3.1), cold light source (Section 3.2), NIR laser (Section 3.3), FUS (Section 3.4), and capacitive hyperthermia (Section 3.5) methods.

**Figure 2 cancers-13-04628-f002:**
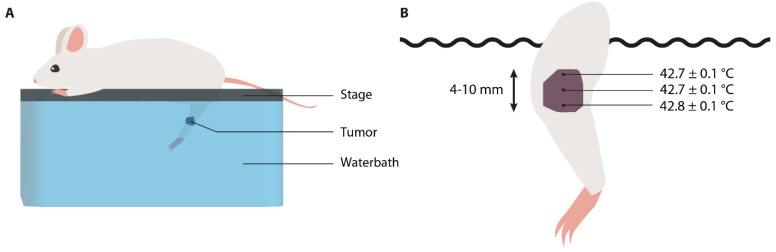
Illustrations of a water bath unit for hyperthermia application. (**A**) Immersion of the target region in a temperature-controlled water bath; (**B**) the intratumoral temperature distribution of a tumor-bearing leg immersed (43 °C, 5 min). The aligned temperature probes were placed in the tumor center at different depths in relation to the water surface. Image B is adapted from Nishimura et al. [29].

**Figure 4 cancers-13-04628-f004:**
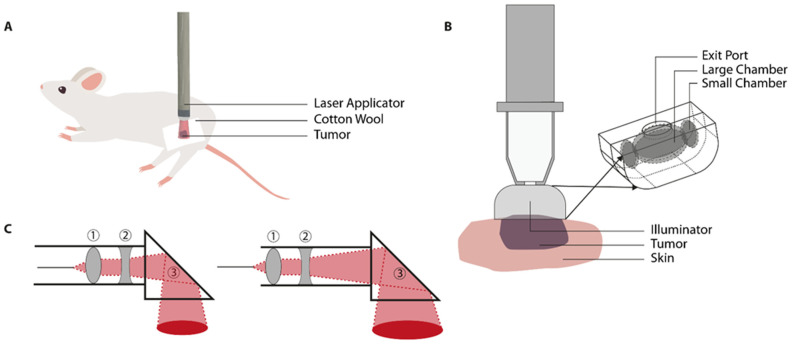
Illustrations of the near-infrared laser light device for hyperthermia application. (**A**) This system is based on a single laser applicator. The healthy tissues can be shielded either by cotton wool or through the use of an illuminator; (**B**) uniform laser light can be applied through the use of an illuminator. By passing the light through the chambers, reflections result in spatially homogenized light exiting the device; (**C**) laser beam adjustments can also be made through the use of two lenses and one prism. By moving the lenses (1,2) further from the prism (3) the beam width expands. Image B is adapted from Dou et al. [101] and image C is adapted from Willerding et al. [41].

**Table 6 cancers-13-04628-t006:** Overview of the basic hyperthermia devices for preclinical small animal studies.

	Water Bath	Cold-Light Source	Near-Infrared Laser	Focused Ultrasound	Capacitive Hyperthermia
**Method**	Tissue heating via convection and conduction by immersion in water	Visible Light via halogen or wIRA device	High energy laser light	Ultrasound waves heat tissue noninvasively via transducer	Capacitive heating though generation of an electromagnetic field
**Tumor** **Position**	Superficial (Flank, Breast, or Hind Limb)	Superficial (Flank or Hind Limb)	Superficial (Flank, Hind Limb, or Mammary Fat Pad)	Superficial and Deep (Flank, Abdominal Wall, or Pancreas)	Superficial and Deep (Flank, Mammary Fat Pad, or Prostate)
**Setting**	Water bath temperature	350–1180 nm	763–1064 nm	1.00–1.54 MHz	8.00 or 13.56 MHz
**Device**	Water temperature (°C) Treatment Duration (s)	Wavelength (λ) No. of Light Guides Lamp Intensity (W/cm^2^) Treatment Duration (s)	Wavelength (λ) Adjust beam size (mm) Laser Intensity (W/cm^2^) Pulse Interval (s) Treatment Duration (s)	Adjust beam size and geometry (mm) Acoustic Intensity (W/cm^2^) Duty Cycle (%) Pulse Repetition Frequency (Hz) Treatment Duration (s)	Adjust electrode size (mm) Power (W) Treatment duration (s)
**Positioning**	TESA tape Jig Sinker	Vet wrap	-	Custom holder	-
**Protection**	Skin	Skin	Skin	Skin	Skin
	Vaseline Cream Plastic Bag (Wetted) Gauze	Vaseline Cream Gauze Polystyrene Aluminum foil	Cotton Wool Polystyrene	Depilation Coupling Medium	Depilation Coupling Medium Dampened Gauze
	Stage	Stage	Stage	Stage	Stage
	Polystyrene Plastic	-	-	Acoustic Absorber	Conductive material Temperature-controlled
**Side-Effects**	Erythema Hemorrhagic Necrosis	Erythema	Skin damage; tissue dehydration and charring	Indirect skin damage Far-field heating	Heat absorption adipose tissue
**Advantages**	Low technical complexity Low cost	Low technical complexity Wavelength tuning	High precision focal zone Homogeneous, deep heating	High precision focal zone	Tumor-specific targeting
**Disadvantages**	Non-specific heating	Limited penetration depth	High technical complexity High cost	High technical complexity High cost	Sensitive to electrical flow disruption (heat deposition)

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
