# Peer review of "External Basic Hyperthermia Devices for Preclinical Studies in Small Animals"

_cancers, 2021, doi:10.3390/cancers13184628_

Round 1

Reviewer 1 Report

The  authors are congratulated for their attempts to review various methods of heating small animal tumors.  However, unfortunately, in discussing various methods of heating, the dept of detail varies considerably among different heating devices and lack of consistency. For   example, whereas there are many capacitive heating methods and devices, only one machine/method ( (mEHT) is discussed in detail.  Redundant discussions are scattered through the manuscript. In discussing  HIFU, it is sated that “the  applied thermal dose is of importance, which is dependent on the applied maximum  temperature and the treatment during. It  should. --------- can last upto 24 h (Line 431-433). This conclusion is true regardless of heating method and not only for the HIFU heating.

Reviewer 2 Report

This review provides a comprehensive overview of preclinical systems for solid tumor hyperthermia treatment in small animals. The paper addresses an unmet need of great relevance to the hyperthermia community, as there is no review for preclinical hyperthermia devices. The secondary goal of promoting standardization is much welcomed. There are no major reviews required, except for the lack of microwave devices that have been reported in at least one preclinical study by Salahi et al 2012. https://pubmed.ncbi.nlm.nih.gov/22690856/. Your group has also developed a MW mouse applicator that also deserves to be mentioned: Raaijmakers et al. 2018 https://pubmed.ncbi.nlm.nih.gov/28828891/ in a microwave section or at the very least a paragraph.

You will find 59 comments in the attached pdf and I expect all to be addressed. No need to comment back on prescriptive changes. I summarized below the most important ones.

English

  1. Overall, it is well written. However, some terms lack adjectives, some paragraphs appear to be disconnected from the surrounding ones, and there are moments where parameter ranges would enrich the reader experience; not to mention sparing the reader the extra work of searching for such ranges. Also, “set-up” is incorrect: please use “setup”.
  2. Either use space between units and values or don’t. Please be consistent.

Structure:

  1. I see value in separating the section “2.2. Anesthesia and physiological monitoring” into 2 different ones.

Content:

  1. The effect of the tail in murine thermoregulation was omitted. I recommend you mention it the anesthesia/thermoregulation section. Since one of the goals of the review is to aim at standardization, any prospective researcher should be aware of the role of the tail in thermoregulation.

CONCLUSIONS

  1. Very good conclusions, except for the end which is quite weak. Incredibly nonspecific. Please make it relevant to this review paper.

FIGURES:

  1. Please increase label font size and add labels as indicated in the pdf.

REFERENCES

  1. Fix “Error! Reference source not found.” that shows up several times in the text.

VERDICT

I recommend this paper to be accepted after addressing the above-summarized comments as well as the ones detailed in the attached pdf.

Author Response

This manuscript is a resubmission of an earlier submission. The following is a list of the peer review reports and author responses from that submission.

Round 1

Reviewer 1 Report

The paper is a structured review of hyperthermia devices dedicated for small animals (mouse, hamsters and rats) and would be of great interest for hyperthermia researchers designing animal studies. However, before it can be published, this paper needs a lot of work. Although the main hyperthermia devices for small animals are mentioned, I consider it a missed opportunity to write a systematic review. Also, the connection with the hyperthermia clinic, where mostly electromagnetic equipment is used is lacking. A thorough discussion when to use which hyperthermia device to heat an animal in which study setup for a certain tumor type would be of great additional value. A discussion comparing the devices should be added.

Throughout the paper a lot of double spaces are used, please delete them.

Introduction

  1. The introduction is unclear and does not lead to a clear research question.
  2. The last part of the introduction are methods, please rename.
  3. That hyperthermia is an adjuvant treatment is mentioned in different words twice. Please rewrite. Line 30 and line 35-36.
  4. Line 29: Only mild hyperthermia is described, i.e. the “former”, there is no “latter”, why would you mention ablation? Or describe ablation in more detail.
  5. I would strongly advise to write a systematic review.
  6. Why only Pubmed?
  7. What were the exact search terms?
  8. Please provide an overview how many papers were found with the search, duplicates, how many papers were excluded based on the titles, abstracts, etc. Use PRISMA.
  9. Who did the search, exclusion etc?
  10. How many articles were not found through you search (key publications, additional material), in other words, was your search complete? In a systematic review also unpublished data is included (f.e. abstracts), in this review the small animal device under development by ALBA is not mentioned.
  11. Line 53 “integrating hyperthermia”, replace by “applying hyperthermia”?

Hyperthermia devices - General considerations

Although I understand why the authors have chosen to start with the tumor model, it would be more clear when a short summary of the results of the search are given first.

  1. Start with the results of your search, and a PRISMA figure.
  2. Then an introduction paragraph naming the different techniques and that they are described in more detail later, use table 5.
  3. Line 69, references 22-24 are named earlier than reference 13 (line 78).
  4. Line 78 “more prominent” rename as “typical”.
  5. Line 80-82, resulting in ?
  6. Line 85-86, depends on endpoint.
  7. Line 91, the point of this sentence is to name the typical tumor sizes, then why are the tumor sizes between brackets?
  8. Line 93, “naturally” is not correct.
  9. Report that hamsters are only used in 5 studies.
  10. Why should you choose a mouse, hamster or rat when designing a small animal hyperthermia study?

Anesthesia and physicological monitoring

  1. Start every paragraph with a short introduction instead of in depth.
  2. Line 95, “is applied”, is this always the case, of in all of the papers?
  3. In this first paragraph references should be added to each statement.
  4. Line 128 to 144 is important information for hyperthermia studies. Please put this in a separate paragraph.

Reviewer 2 Report

The authors are congratulated for their extensive review on the experimental hyperthermia using various tumor models and heating devices.  Some fundamental principles of hyperthermia are addressed and various heating devices are described. Unfortunately, however, there are numerous problems in this manuscript including use of poor English language and inaccurate statements on various aspects of hyperthermia and experimental procedures. “ Is”  “was” “ will “ are incorrectly used throughout the manuscript.

Following are few examples of problems identified.

For the induction of tumors in experimental animals, either suspension of tumor cells or fragment of tumor tissues are inoculated. However, it is stated at many places in this review paper that  tumors are induced by inoculating tumor fragment; inoculation of single cell suspension is a common practice in inducing experimental tumors in rodents.

Followings are few problems identified.

L 28.   ---------ranging from 40-43oC ----.  -----ranging from 40 to 43oC --      or in the range of 40-43oC.

L35.   -----------have in common that they are---. Awkward

L46.  -----------  in sight in the ---.  ----insight in to the ---

 L 52 ------------ from  a slight increase in physiological temperature.    from  a slight increase in tumor  temperature.    

L77.     ------- tumor  growth speed. In tumor biology, we usually say tumor  growth rate and not speed.

L84. ------influence  the efficacy ( of what ?).    ----influence the efficacy of heating .

L92.  Overall, all of these traits -------L120   :  The meaning of this statement is unclear.

L 120. ---both tumor skin surface- and deep  --.  What do you mean by “ tumor skin surface” ?

L 145.  Hyperthermia Devices-without optical monitoring.    Why is without optical monitoring needed here ?

L148.  Water  bath heating “ is considered to be --- “  This review think it is and not “ is considered”

L178.  ---as it is only suitable for model studying superficial tumors.   Water bath heating is frequently used for heating subcutaneous or intramuscular tumors.

L 184.  Not the hostile micro-environment has created a poor circulation but the poor circulation creates hostile micro-environment in tumors.

L 204.  ---this value will decrease ------.  --- temperature  decreases  ---
